# Estimating Commonsense Plausibility through Semantic Shifts

## Abstract

Commonsense plausibility estimation is critical for evaluating language models (LMs), yet existing generative approaches–reliant on likelihoods or verbalized judgments–struggle with fine-grained discrimination. In this paper, we propose ComPaSS, a novel discriminative framework that quantifies commonsense plausibility by measuring semantic shifts when augmenting sentences with commonsense-related information. Plausible augmentations induce minimal shifts in semantics, while implausible ones result in substantial deviations. Evaluations on two types of fine-grained commonsense plausibility estimation tasks across varying input formats and commonsense knowledge levels based on different backbones, including LLMs and vision-language models (VLMs), show that ComPaSS consistently outperforms baselines. It demonstrates the advantage of discriminative approaches over generative methods in fine-grained commonsense plausibility evaluation. Experiments also show that (1) VLMs yield superior performance to LMs, when integrated with ComPaSS, on vision-grounded commonsense tasks. (2) contrastive pre-training sharpens backbone models' ability to capture semantic nuances, thereby further enhancing ComPaSS.

## 1 Introduction

Commonsense knowledge–the shared understanding of everyday phenomena and human experiences Schank (1983); Winograd (1986); Hobbs (1990)–is foundational to natural language understanding and generation. Despite the remarkable progress in large language models' (LLMs) text generation capabilities, ensuring commonsense plausibility in their outputs remains an unresolved challenge Marcus (2020); Elazar et al. (2021); Mahowald et al. (2024); Chen et al. (2023). This challenge arises not only from the inherent difficulty of acquiring and applying commonsense knowledge but also from the absence of reliable frameworks for evaluating textual plausibility. Effective evaluation of commonsense plausibility addresses this gap twofold: it identifies commonsense violations Miranda et al. (2024); Saravanan et al. (2024) while offering quantifiable metrics to guide the development of techniques that augment LLM outputs Tian et al. (2023).

In this work, we focus on developing generalizable methods for commonsense plausibility estimation (CSPE) that can be applied across diverse domains and tasks. This leads us to investigate zero-shot and few-shot approaches based on pre-trained LMs, which leverage their inherent knowledge without requiring additional training data or domain-specific fine-tuning.

Previous studies on zero or few-shot CSPE primarily adopt a generative perspective and can be categorized into two main approaches, likelihood estimation and verbalized judgments. The likelihood-based methods Trinh & Le (2018); Tamborrino et al. (2020); Holtzman et al. (2021) utilize token prediction probabilities from language models as an indicator, with the assumption that sentences consistent with commonsense knowledge tend to have a higher likelihood for their component tokens. The verbalization-based methods Brown et al. (2020); Krause & Stolzenburg (2024) ask pre-trained LMs to answer the plausibility of a sentence through natural language. The models can generate the answer based on knowledge stored in their parameters.

However, approaches based on the generative perspective could be suboptimal for CSPE, since it is essentially a discriminative task. In this paper, we adopt a discriminative perspective for CSPE. In communication, commonsense knowledge is often assumed and left unstated, yet such omissions rarely hinder mutual understanding Clark (1996); Noveck & Sperber (2004). Inspired by this, we

propose ComPaSS, a method that measures **Com**monsense **Pl**ausibility through **S**emantic **S**hifts introduced when augmenting sentences with commonsense-related information. Plausible additions yield minimal semantic shifts, whereas implausible ones result in substantial deviations. For instance, adding 'black' to 'There is a penguin' results in a minor semantic shift, aligning with the penguins' natural coloration. By contrast, introducing 'green' creates a substantial shift, highlighting the implausibility of such an atypical attribute. To quantify semantic shifts, ComPaSS computes the similarity between embeddings of the original sentence (without explicit commonsense references) and its modified counterpart augmented with commonsense-related information.

Two aspects of semantic representations could influence the capability of ComPaSS in CSPE: the inclusion of commonsense knowledge and the discrimination of semantic nuances. These correspond to two key aspects of models used for obtaining sentence embeddings: 1) Modality. Language Models (LMs) often suffer from *reporting bias* Gordon & Durme (2013), which involves systematic distortions due to omitted commonsense details (e.g., 'penguins are black' is rarely stated) and statistical biases from fixed linguistic patterns (e.g., 'black sheep'). In contrast, vision-language models (VLMs) incorporate visual information, thus mitigating reporting bias, especially for visually-grounded commonsense knowledge (e.g., object colors or spatial relations) Paik et al. (2021); Zhang et al. (2022). 2) Contrastive learning. By training a model to distinguish between semantically similar and dissimilar instances, it enhances the model's discriminative power. Representations from contrastively trained models exhibit sharper separability, which directly impacts the precision of semantic shift measurements. Given these considerations, we study how ComPaSS performs based on various backbones of both LMs and VLMs, with and without contrastive learning.

We evaluate ComPaSS against baselines on two fine-grained CSPE tasks that require ranking candidate answers by plausibility rather than binary classification. These tasks prioritize nuanced plausibility judgments, where answers may hold varying degrees of validity. The first task, attribute value ranking (CoDa Paik et al. (2021) and ViComTe Zhang et al. (2022)), involves ranking candidate attribute values (e.g., color, shape, material) for objects using structured triplets as input (e.g., determining that "black" is more plausible than "green" for penguin-color). The second task, commonsense frame completion Cheng et al. (2024), challenges models to rank plausible completions for free-form open-ended questions (e.g., selecting 'farm' over 'truck' for 'Where are farmers with newly harvested crops?'), testing alignment with human preferences and broader commonsense reasoning. Together, these tasks assess ComPaSS across input formats (structured triplets vs. free-form text) and knowledge types (object-specific attributes vs. general everyday commonsense).

Our experiments reveal three critical insights. First, as a discriminative approach, ComPaSS consistently outperforms prior generative methods in fine-grained plausibility estimation, achieving superior results across diverse model backbones. This highlights the advantage of discriminative methods in capturing subtle plausibility distinctions. Second, utilizing ComPaSS, VLMs significantly outperform LMs for vision-grounded commonsense (e.g., object colors or shapes), demonstrating that visual information enhances representations and benefits CSPE. Third, models with contrastive pre-training yield significantly better results than those without, emphasizing the importance of representations that capture semantic nuances in plausibility measurement through ComPaSS.

## 2 RELATED WORK

### 2.1 CSPE BASED ON INTERNAL KNOWLEDGE

The sentence probability and perplexity computed by LMs can serve as indicators of commonsense plausibility, even in zero-shot settings Trinh & Le (2018); Davison et al. (2019); Liu et al. (2021a). For LLMs with instruction-following capability, they can be directly prompted to judge whether a given input is consistent with commonsense or not Zhao et al. (2024). Beyond directly judging plausibility, some methods Jung et al. (2022); Tafjord et al. (2022) evaluate the plausibility of hypotheses by scoring the validity of entailment paths generated by the LLMs, i.e., the reasoning chains justifying 'reasonable' or 'unreasonable' conclusions, and selecting the final prediction based on the highest-scoring path. VERA Liu et al. (2023) adopts a discriminative approach, training a classification head to make predictions based on model representations, which fine-tunes LLMs on ~7 million commonsense statements. In contrast, our approach also leverages internal knowledge from a discriminative perspective but does not require additional training.

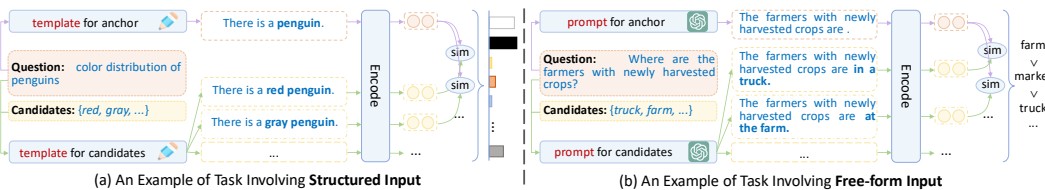

Figure 1: How ComPaSS works on different tasks.

## 2.2 CSPE BASED ON EXTERNAL KNOWLEDGE

Language models (LMs) may have insufficient or inaccurate knowledge, which led to some methods to incorporate external knowledge to better estimate commonsense plausibility. A typical approach is to augment the model's knowledge by retrieving relevant sentences from external sources Zhang et al. (2021); Yu et al. (2022). Commonsense knowledge bases (KBs) Speer et al. (2016); Sap et al. (2019); Hwang et al. (2020) store extensive commonsense knowledge, enabling the extraction of relevant subgraphs to evaluate sentence consistency with commonsense Choi et al. (2022). To alleviate the coverage limitations of the KBs while leveraging the extensive knowledge encoded in LMs, COMET Bosselut et al. (2019) introduced a dynamic KB by pre-training LM on existing commonsense KBs. Methods that utilize this dynamic KB Ghazarian et al. (2023); Tian et al. (2023) demonstrate improved generalization across various commonsense reasoning tasks.

## 3 TASK DEFINITION

Formally, given an input instance $x_i = (c; a_i^c)$ consisting of a context $c$ and a candidate information $a_i^c \in A$, where $A^c = \{a_1^c, a_2^c, ..., a_K^c\}$ denotes the context-dependent candidate set with size $K$, the task is to predict a plausibility score set $\mathcal{P}^c = \{p_1^c, p_2^c, ..., p_K^c\}$ for all candidates, where each $p_i^c \in \mathbb{R}$ quantifies the plausibility of augmenting $c$ with $a_i^c$. The ground-truth scores are denoted as $\mathcal{G}^c = \{g_1^c, g_2^c, ..., g_K^c\}$, where $g_i^c$ indicates the true score of $a_i^c$. Performance is measured by the correlation between $\mathcal{P}^c$ and $\mathcal{G}^c$.

The input can take two specific forms: for *attribute value ranking* task, the input is a structured triplet $x_i = (o, \text{has property } p; a_i^c)$. The context $c = (o, \text{has property } p)$, where $o$ is a common object and $p$ is a property. The candidate $a_i^c$ represents the $i$-th attribute value for the specified property. For the *commonsense frame completion* task, the context $c = q$ is a free-form question, the input is a question-answer pair $x_i = (q; a_i^c)$, where $a_i^c$ is the $i$-th plausible answer to this question.

## 4 COMPASS

Our method, ComPaSS, is a zero-shot approach for estimating commonsense plausibility. We demonstrate in Figure 1 how this method works on different tasks. For each input, we first construct an anchor sentence (omitting the commonsense-related detail) and a candidate sentence (augmenting that detail). We then encode both sentences individually to obtain their semantic representations. Next, we calculate their semantic similarity, where the degree of semantic shift—inversely proportional to similarity—quantifies plausibility.

### 4.1 CONSTRUCTING SENTENCES

For each input context $c$ and the candidate to be evaluated $a_i^c$, we construct two types of sentences: an anchor sentence $s_{\text{anchor}}$ that contains only the base context $c$ while omitting target details, and a candidate sentence $s_{\text{candi}}$ that further incorporates commonsense-related information $a_i^c$. The construction process varies based on input type but follows a unified framework:

$$s_{\text{anchor}} = f_{\text{anchor}}(c, z_{\text{anchor}}), \tag{1}$$

$$s_{\text{candi}} = f_{\text{candi}}(c, a_i^c, z_{\text{candi}}), \tag{2}$$

where $f(\cdot) \in \{f_{\text{anchor}}(\cdot), f_{\text{candi}}(\cdot)\}$ denotes the construction function, and $z \in \{z_{\text{anchor}}, z_{\text{candi}}\}$ denotes task-specific templates or prompts.

As illustrated in Figure 1, the framework is instantiated differently based on the input format: For *structured triplet inputs*, we employ template-based construction, where $z$ represents a pre-defined template (see Appendix D) and $f(\cdot)$ represents applying this template to generate a sentence. In contrast, for tasks involving *free-form question-answer pairs as input*, we query GPT-4 Achiam et al. (2023) to generate contextually coherent sentences, where $z$ denotes the prompt (see Appendix C) and $f(\cdot)$ represents querying GPT-4 using the prompt. Since questions cannot be directly converted into coherent statements, we use a blank space as a placeholder when constructing anchor sentences. Such an adaptive construction method enables ComPaSS to be applicable to different input forms.

## 4.2 Representing Sentences

Given anchor and candidate sentences, we encode them into dense semantic representations using a pre-trained model $\theta$, which can be either a LM or a VLM. For each sentence $s \in \{s_{\text{anchor}}, s_{\text{candi}}\}$, the model first processes the sentence along with special tokens (e.g., [CLS], [EOS], or others depending on the model architecture) and then outputs token hidden states:

$$H = \theta(s) = \{h_0, h_1, ..., h_l\}, \tag{3}$$

where $l$ denotes the sequence length, including the special tokens. The final sentence representation $r \in \{r_{\text{anchor}}, r_{\text{candi}}\}$ is derived through architecture-specific strategies.

For encoder models, we use the hidden state of the designated semantic aggregation token as sentence representation. Some models (e.g., RoBERTa Liu et al. (2021b)) use the initial '[CLS]' token for sentence representation ($r = h_0$), while others (e.g., CLIP Radford et al. (2021)) utilize the final '[EOS]' token embedding ($r = h_l$).

For decoder models, we use the hidden state of the last token as sentence representation $r = h_l$, which naturally encapsulates the accumulated context. Alternatively, PromptReps Zhuang et al. (2024) prompts the model to generate a new representative token at position $l + 1$, using its hidden state as the sentence representation ($r = h_{l+1}$). We apply this strategy to models that are not enhanced by contrastive learning.

This architecture-aware representation strategy ensures ComPaSS's flexibility across different model backbones while maintaining optimal performance for each specific architecture.

## 4.3 Ranking with Semantic Shifts

We rank the candidate option $a_i^c$ by measuring how naturally it integrates into the context, quantified through semantic similarity between the anchor sentence representation $r_{\text{anchor}}$ and the candidate sentence representation $r_{\text{candi}}$. The underlying principle is that the more plausible the information, the smaller the semantic shifts it induces when added to the context, leading to higher semantic similarity. Formally, we define the commonsense plausibility score $p_i^c$ for each candidate $a_i^c$ as:

$$p_i^c \propto \text{sim}(r_{\text{anchor}}, r_{\text{candi}}), \tag{4}$$

where $\text{sim}(\cdot)$ denotes a similarity function (e.g., cosine similarity or dot product). Candidates are then ranked by their plausibility scores descendingly, with higher-ranked candidates representing more commonsense-consistent answers.

## 4.4 Discussion of Applicable LMs

This paragraph discusses the differences in applicable LMs between ComPaSS and generative methods based on likelihoods and verbalization. ComPaSS can utilize both encoder and decoder models as long as they can yield reasonable sentence representations. Likelihood-based approaches can also leverage these two types of LMs. Candidate likelihoods can be estimated based on masked/next token prediction for encoders and decoders respectively. In contrast, verbalization-based approaches require LLMs–decoder-only LMs–to answer the plausibility estimation questions. This indicates the broader applicability of ComPaSS.

## 5 EXPERIMENTAL SETUP

### 5.1 DATASETS

We evaluate methods through two types of fine-grained commonsense plausibility estimation (CSPE) tasks, where candidates should be ranked based on commonsense plausibility. These tasks are chosen to comprehensively evaluate methods across varying input formats (from structured triplets to free-form text) and commonsense knowledge levels (from specific attribute knowledge to general everyday commonsense knowledge).

#### 5.1.1 STRUCTURED ATTRIBUTE KNOWLEDGE

**Color Dataset (CoDa)** [1] Paik et al. (2021) is a human-annotated dataset used for attribute value ranking, which provides color distributions for commonly recognized objects. It contains 521 objects, each with 11 candidate color attributes.

**Visual Commonsense Tests (ViComTe)** [2] Zhang et al. (2022) is another dataset used for attribute value ranking, which is derived from Visual Genome Krishna et al. (2017). It offers attribute value distributions across broader properties, including color, shape, and material. It contains 2,877 objects with 12 candidate color attributes, 706 objects with 12 candidate shape attributes, and 1,423 objects with 18 candidate material attributes.

#### 5.1.2 FREE-FORM GENERAL KNOWLEDGE

**Commonsense Frame Completion (CFC)** [3] Cheng et al. (2024) is a dataset designed to evaluate implicit commonsense reasoning, which consists of questions accompanied by multiple plausible answers with human-annotated preference scores. It requires models to make probabilistic judgments about answer plausibility, which should align with human preferences. As the test set is not public, we use the validation set containing 55 questions for zero-shot evaluation.

### 5.2 EVALUATION METRICS

**Spearman's rank correlation coefficient** $\rho$: We choose this as the primary metric following CoDa and ViComTe. It measures the rank correlation between predicted and ground-truth plausibility orderings. This emphasis on relative ordering aligns with the nature of commonsense plausibility assessment, where the exact probability values are less important than correctly identifying more plausible options over less plausible ones.

**Accuracy**: CoDa and ViComTe also include binary comparison tasks where each object is paired with two attribute values, with one more plausible than the other. Models need to rank the more plausible value higher. Accuracy quantifies the success rate of these binary selections. This metric is suitable for cross-attribute comparisons as it is unaffected by variations in the number of candidates, unlike Spearman's rank correlation coefficient.

### 5.3 METHODS FOR COMPARISON

#### 5.3.1 COMPASS WITH VARIOUS BACKBONES

We evaluate ComPaSS across diverse model architectures to assess its adaptability:

For LMs, we evaluate both base models and their contrastive learning pre-trained variants: RoBERTa-Large Liu et al. (2021b) (RoBERTa) is a widely-used encoder-only LM with fewer parameters. Mistral-7B-Instruct Jiang et al. (2023) (Mistral) and Qwen2-7B-instruct qwe (2024) (Qwen2) are two decoder-only LLMs with strong instruction-following capabilities. We also evaluate their **contrastive learning pre-trained** variants, i.e., sup-SimCSE-RoBERTa-Large Gao et al. (2021) (RoBERTa$_{w/CL}$), E5-Mistral-7B-Instruct Wang et al. (2023; 2022) (Mistral$_{w/CL}$) and gte-Qwen2-7B-instruct Li et al. (2023) (Qwen2$_{w/CL}$). Please note that all contrastive learning procedures are

---

[1] https://github.com/nala-cub/coda

[2] https://github.com/ChenyuHeidiZhang/VL-commonsense

[3] https://github.com/qxc101/PROBEVAL_CFC/

| Model (#Inference Parameters) | CoDa | Color | Shape | Material | CFC |
|---|---|---|---|---|---|
| Baselines | | | | | |
| **CSM** ACCENT (440M) | 10.07 | 10.35 | -2.10 | 16.99 | 35.04 |
| COMET-Atomic (440M) | 22.91 | 26.98 | 40.44 | 25.72 | - |
| VERA-T5 (5B) | 58.93 | 45.08 | 30.31 | 33.51 | 45.81 |
| **LM** RoBERTa+likelihood (355M) | 24.37 | 33.63 | 36.12 | 24.23 | 42.46 |
| RoBERTa$_{w/CL}$+likelihood (355M) | 23.36 | 31.51 | 26.69 | 22.23 | 38.03 |
| Mistral+verbal. (7B) | 46.64 | 38.63 | 30.46 | 36.34 | 32.06 |
| Mistral+likelihood (7B) | 51.30 | 34.31 | 26.70 | 37.03 | 47.98 |
| Qwen2+verbal. (7B) | 57.40 | 41.59 | 38.30 | 36.76 | 29.32 |
| Qwen2+likelihood (7B) | 50.25 | 40.99 | 32.52 | 37.13 | 45.10 |
| Qwen2$_{w/CL}$+likelihood (7B) | 49.65 | 41.75 | 32.80 | 37.30 | 43.00 |
| ComPASS | | | | | |
| **LM** RoBERTa$_{w/CL}$ (355M) | 44.59 | 38.92 | 42.92 | 33.55 | 44.46 |
| Mistral$_{w/CL}$ (7B) | 58.54 | 42.20 | 43.75 | **38.77** | **49.01** |
| Qwen2$_{w/CL}$ (7B) | 59.16 | 44.61 | 47.51 | 38.49 | 46.41 |
| **VLM** CLIP (124M) | 58.10 | 45.55 | 45.82 | 33.56 | 35.13 |
| EVA-CLIP (695M) | **62.87** | **51.73** | **48.05** | 38.67 | 41.46 |

Table 1: Spearman's rank correlation coefficient $\rho$ between the predicted ranks of candidates and their ground-truth on CoDa, ViComTe (Color, Shape, and Material), and CFC, shown in percentage. The **best** and second best results are highlighted in bold and underlined, respectively. '+verbal.' indicates using the verbalization-based method.

pre-training stage optimizations unrelated to our task. We directly use their released checkpoints without task-specific fine-tuning.

For VLMs, we test CLIP-ViT-L/14 Radford et al. (2021) (CLIP), a multimodal representation model trained on image-text pairs using **contrastive learning**, which aligns semantically similar images and text into closely matching representations. We also consider its advanced variant EVA-CLIP-8B Sun et al. (2023) (EVA-CLIP).

### 5.3.2 BASELINES

**Commonsense models (CSMs):** These models are specifically designed for modeling commonsense knowledge: COMET-Atomic-2020-Bart Bosselut et al. (2019) (COME-Atomic) is a commonsense LM pre-trained on commonsense KBs. COMET is suitable for processing triple input, which can generate a probability score for each candidate. ACCENT Ghazarian et al. (2023) assesses the commonsense plausibility of a sentence by first extracting structured tuples and then scoring them based on their compatibility with a commonsense KB. VERA-T5-XXL Liu et al. (2023) (VERA-T5) is trained on ~7M commonsense statements and can directly estimate the commonsense plausibility of statements.

**Language models (LMs):** We evaluate all open-source LMs used as the backbone of ComPaSS with two methods. For the *likelihood based* method, the plausibility of a sentence is proportional to the normalized probability of predicting each token sequentially. For the *verbalization based* method, pre-trained LMs are prompted in natural language (see Appendix B) to rank candidates based on plausibility. We also test closed-source LLMs including gpt-3.5-turbo-0125 OpenAI (2022) (GPT-3.5) and gpt-4-0125-preview Achiam et al. (2023) (GPT-4), the latter introduces multimodal technology with superior capabilities.

### 5.4 IMPLEMENTATION DETAILS

All experiments are carried out in a zero-shot or in-context few-shot setting. Closed-source models are accessed via official APIs, while open-source implementations run on a single NVIDIA A800 80G GPU. For ACCENT, the beam number is 10 as the official setting. When testing the CFC

| Method | CoDa | Color | Shape | Material |
|---|---|---|---|---|
| likelihood | 24.37 | 33.63 | 36.12 | 24.23 |
| ComPaSS | 24.63 | 22.68 | 26.77 | 19.93 |
| w/ unsup-CL | 32.67 | 32.00 | 42.18 | 31.12 |
| w/ sup-CL | 44.59 | 38.92 | 42.92 | 33.55 |

Table 2: Performance of different Roberta variants. By default we use the vanilla RoBERTa. 'w/ unsup-CL' and 'w/ sup-CL' denote RoBERTa pre-trained with unsupervised and supervised contrastive learning, respectively.

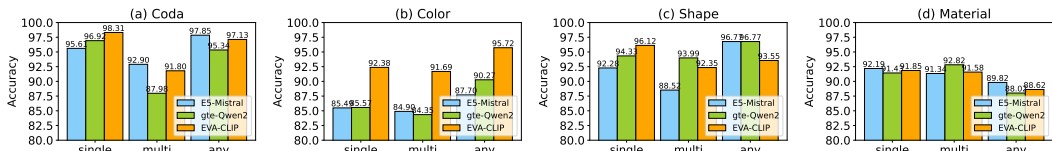

Figure 2: Binary classification accuracy of models with ComPaSS on different groups.

dataset using the verbalization method, we sample the model 100 times for each question with a temperature of 0.7, and cluster answers follow the official protocol.

# 6 RESULTS AND ANALYSIS

## 6.1 OVERALL RESULTS

The overall experimental results are presented in Table 1, which reveals several key findings:

**ComPaSS achieves the best performance compared to baselines across both structured triplets (attribute ranking) and free-form text (CFC) inputs.** This demonstrates its robustness to diverse input formats without relying on task-specific templates. Further comparison between RoBERTa, Mistral, and Qwen2, with and without ComPaSS, shows a consistent improvement when ComPaSS is applied. This validates our method's architecture-agnostic effectiveness. Notably, even VERA, which was specifically fine-tuned for CSPE, achieves only comparable performance to ComPaSS-enhanced models. Comparing the performance of different methods on LMs in the baseline, we find that verbalization-based methods fail to consistently outperform likelihood-based approaches, even when applied to generative models. This limitation highlights the challenges such methods face in making fine-grained distinctions required for precise plausibility estimation, whereas Com-PaSS succeeds by unifying semantic shift measurement across both templated and non-templated scenarios.

**VLMs demonstrate superior effectiveness in learning visual-related commonsense knowledge.** Comparing the ComPaSS methods based on various backbones, we find VLMs exhibit particular strength in visual attribute ranking, with EVA-CLIP achieving the highest scores on CoDa (62.87), Color (51.73), and Shape (48.05), significantly outperforming even 7B parameter LLMs. This performance gap persists despite the LLMs' access to large-scale text corpora and additional parameters, underscoring the unique value of visual supervision. This performance gap highlights the limitations of text-only training, as even extensive textual data and additional parameters cannot fully compensate for the lack of visual grounding, which underscores the importance of multimodal learning for comprehensive commonsense understanding.

**Discriminative approaches may offer a more parameter-efficient pathway compared to generative methods.** Our experiments reveal that encoder-only models with millions of parameters like RoBERTa and CLIP-series models achieve comparable or even superior results to much larger decoder-only models (with billions of parameters) when combined with ComPaSS. This suggests that our discriminative method effectively leverages the semantic representation strengths of encoder models, which are generally more parameter-efficient than generative models. By focusing

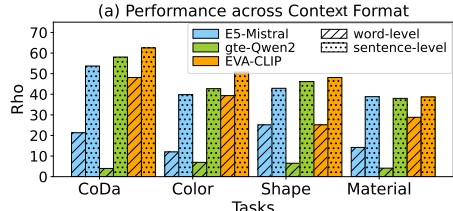 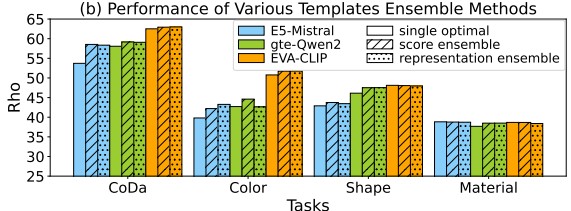

Figure 3: ComPaSS performance with different context formats and ensemble settings.

| Model | CoDa | Color | Shape | Material |
|---|---|---|---|---|
| GPT-3.5 | 94.05 | 92.25 | 90.08 | 89.60 |
| GPT-4 | 94.63 | **93.29** | 89.24 | 88.76 |
| Mistral$_{w/CL}$ | 94.97 | 86.06 | 91.50 | **91.27** |
| Qwen2$_{w/CL}$ | 94.71 | 86.79 | 94.04 | 90.42 |
| EVA-CLIP | **95.39** | **93.29** | **94.33** | 90.79 |

Table 3: Binary comparison accuracy on CoDa and ViComTe. The best results are highlighted in bold. All results are shown in percentage. Both Mistral and EVA-CLIP use the ComPaSS method.

on representation-level semantics rather than token generation, ComPaSS aligns closely with the pre-training objectives of encoder models, maximizing their representation power.

**The ability to discern semantic nuances in sentence representations is crucial for ComPaSS performance.** As shown in Table 2, experiments with different RoBERTa variants reveal that applying ComPaSS to vanilla RoBERTa leads to performance degradation due to its weaker representation capabilities. However, incorporating contrastive learning (even via unsupervised training) significantly improves performance by enabling subtle plausibility distinctions to manifest as measurable embedding space shifts. Crucially, ComPaSS does not require custom contrastive pre-training in practice. It directly leverages contrastively pre-trained SOTA embedding models, enabling continuous performance gains from evolving embedding techniques without task-specific fine-tuning or architectural modifications.

### 6.2 FURTHER ANALYSES

#### 6.2.1 COMPARISONS TO CLOSED-SOURCE MODELS

We extend our evaluation to include state-of-the-art closed-source models, with results presented in Table 3. Notably, our method outperforms even GPT-4 across multiple tasks, demonstrating its effectiveness in fine-grained CSPE. This performance gap further highlights the limitations of verbalization-based approaches in capturing subtle distinctions required for precise plausibility estimation.

#### 6.2.2 GRANULAR ANALYSIS OF ATTRIBUTE TYPES

We analyze binary comparison results on CoDa and ViComTe across three attribute groups: *single*: includes objects with one dominant attribute value (e.g., snow's color), *multi*: includes objects with attributes mainly distributed among the top four values (e.g., a penguin's color), and *any*: includes objects with a broader attribute distribution (e.g., a T-shirt's color). As shown in Figure 3, VLMs demonstrate particular strength in the single group. We attribute this advantage to visual grounding overcoming textual reporting bias: stereotypical attributes are rarely explicitly stated in text due to their commonsense nature, creating a reporting bias in language data. However, these attributes are consistently and explicitly depicted in images, enabling VLMs to overcome linguistic omissions. This finding demonstrates that visual grounding serves as a critical compensator for missing commonsense in text-based training.

| **Task** | : Rank the candidate colors according to the frequency with which a sheep is observed in each color. |
|---|---|
| **Human** | : white, gray, black, brown |
| **GPT-3.5** | : white, black, brown, gray |
| **GPT-4** | : white, black, brown, gray |
| **E5-Mistral\*** | : white, gray, black, brown |
| **get-Qwen2\*** | : white, brown, gray, black |
| **EVA-CLIP\*** | : white, gray, black, brown |

Figure 4: The ranking of sheep colors by humans and different models, along with corresponding images from the physical world (from Google). The '*' in the upper right represents the model with ComPaSS method.

### 6.2.3 EFFECT OF CONTEXT FORMAT

We investigate the importance of sentence-level context in semantic shift measurement by comparing two approaches: *word collocation* comparison (e.g., 'penguin' and 'black penguin') and *full sentence construction* (e.g., 'There is a penguin' and 'There is a black penguin'). As shown in Figure 3(a), sentence-level inputs consistently outperform word-level comparisons for both LLMs and VLMs. This performance gap underscores the importance of complete sentence construction for ComPaSS, as sentence-level inputs better align with models' pre-training data formats.

### 6.2.4 TEMPLATE ENSEMBLE METHODS

For the template-based method, we investigate three ensemble strategies: The *single-optimal ensemble* approach uses the unified best-performing template, serving as an implicit ensemble. For explicit ensemble methods, *score-level ensemble* averages prediction scores across multiple templates, and *representation-level ensemble* fuses sentence representations from several templates before computing the final score. As shown in Figure 3 (b), both explicit ensemble strategies significantly further improve LLM performance, with the score-level ensemble showing more consistent gains. However, VLM shows limited improvement from ensemble methods, likely due to its simpler pre-training data structure. This contrast highlights LLMs' sensitivity to linguistic variations and their ability to benefit from diverse syntactic structures.

### 6.3 CASE STUDY

We use the classic 'black sheep problem' to intuitively explain why ComPaSS is effective. Since 'black sheep' is an idiom, one is much more likely to mention a 'black sheep' than to specify the color of a sheep. Such reporting bias confuses the LMs that learn knowledge through probabilistic modeling. As shown in Figure 4, GPT-3.5 and GPT-4 both overestimate the probability of 'black' being the color of sheep even though sheep in black are rare. In contrast, our approach relies on semantic rather than probabilistic likelihood is able to distinguish between the linguistic meaning and the visual recognition of 'a black sheep', resulting in a more accurate estimation of the sheep's color. In addition, VLM calibrates the color distribution well by incorporating visual information.

## 7 CONCLUSION

We introduce ComPaSS, a discriminative framework for fine-grained commonsense plausibility estimation via semantic shift measurement. By leveraging the idea that plausible commonsense augmentations cause minimal semantic deviation, ComPaSS offers a generalizable approach for various tasks and model architectures. Our experiments show that discriminative methods outperform generative approaches in capturing nuanced plausibility distinctions, with ComPaSS consistently surpassing likelihood-based and verbalization-based baselines. Vision-language models also excel on visually-grounded commonsense tasks, addressing reporting bias through multimodal alignment. Finally, we emphasize the role of contrastive pre-training in improving semantic representation quality, directly enhancing plausibility estimation accuracy. Overall, ComPaSS highlights the value of utilizing semantic embeddings to extract commonsense knowledge from pre-trained models.

## 8 ETHICAL CONSIDERATIONS

As our method relies on LLMs and VLMs, it inherits potential biases present in the training data. These biases, whether related to societal stereotypes or uneven distribution of information across certain attributes, could affect the model's judgment in ranking attribute plausibility. Consequently, our method may inadvertently perpetuate or amplify these biases, especially in scenarios where the model's understanding of an attribute is skewed by biased representations in the data. Addressing these biases is an important avenue for future work.

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

## A  APPENDIX

## B  PROMPT FOR VERBALIZATION-BASED METHOD

The prompt we use for the verbalization-based method can be found in Figure 5.

---

**The Prompt of Verbalization-based Method for Attribute Value Ranking**

Sort all the <PROP>s in candidate set based on how frequently the object is observed to be each <PROP>. The higher the <PROP> is ranked, the more commonly the object is of that <PROP>. The candidate set is <PROP_LIST>. The output must be a sorted result that includes all candidate <PROP>s as in the example.

Here are some samples:
object: <OBJ0>
result: <RES0>
object: <OBJ1>
result: <RES1>
object: <OBJ2>
result: <RES2>

New Task:
object: <OBJ>
result:

---

**The Prompt of Verbalization-based Method for Commonsense Frame Completion**

Answer the question based on commonsense. Your answer should be brief. You cannot refuse to answer for any reason.

Example 1:
Question: who was driving through the night, shooting blurred lights out of focus?
Answer: person
Example 2:
Question: why would an aircraft receive fuel from a cargo aircraft?
Answer: to fly
Example 3:
Question: where's the heart-shaped hot dog and some pizza on a big tray?
Answer: restaurant

New Task:
Question: <Q>
Your answer:

---

Figure 5: The prompt for attribute value ranking task and commonsense frame completion task.

## C  PROMPT FOR SENTENCE TRANSFORMATION

The prompt we use for converting question-answer pair can be found in Figure 6. For the Commonsense Frame Completion (CFC) task, answers with similar semantics (e.g., "person" vs. "a person")

will be further grouped into equivalence clusters during evaluation rather than being considered as individual answers. Following the dataset's official protocol, each question is asked multiple times to estimate the sampling probability of the model as accurately as possible, and different expressions of the same type of answer are allowed to avoid the influence of vocabulary selection on the model.

---

Transform the problem into declarative sentence based on each answer with minimal modifications. Do not introduce more information, and do not lose any information in the questions and answers.

For Example:
Question 1:
who was driving through the night, shooting blurred lights out of focus?
Answers 1:
1. person, 2. chauffeur, 3. taxi driver, 4. a person, 5. or a driver.
Sentences 1:
1. A person was driving through the night, shooting blurred lights out of focus.
2. A chauffeur was driving through the night, shooting blurred lights out of focus.
3. A taxi driver was driving through the night, shooting blurred lights out of focus.
4. A person was driving through the night, shooting blurred lights out of focus.
5. A driver was driving through the night, shooting blurred lights out of focus.
Question 2:
why would a goat eat hay in a stable?
Answers 2:
1. gain energy, 2. to fulfill hunger, 3. to get nutrition, 4. get nutrition
Sentences 2:
1. a goat eats hay in a stable to gain energy.
2. a goat eats hay in a stable to fulfill hunger.
3. a goat eats hay in a stable to get nutrition.
4. a goat eats hay in a stable to get nutrition.
Question 3:
why would an aircraft receive fuel from a cargo aircraft?
Answers 3:
1. longer flight times, 2. takeoff, 3. traveling, 4. enable travel, 5. refill fuel
Sentences 3:
1. an aircraft receives fuel from a cargo aircraft because of longer flight times.
2. an aircraft receives fuel from a cargo aircraft for takeoff.
3. an aircraft receives fuel from a cargo aircraft for traveling.
4. an aircraft receives fuel from a cargo aircraft to enable travel.
5. an aircraft receives fuel from a cargo aircraft to refill fuel.

New Task:
Question 4:
<Q>
Answers 4:
<A>
Sentences 4:

---

Figure 6: The prompt for converting question-answer pair into sentence. The blue part is the instruction, the green part is the 3-shot example, and the red part is the placeholder for the specific input.

## D TEMPLATES FOR SENTENCE CONSTRUCTION

The templates we used to construct anchor sentences and candidate sentences of different property are shown in Table D.

| Property | Templates for anchor | Templates for candidate |
|---|---|---|
| Color | A photo of a [o]. | A photo of a [c] [o]. |
| | A picture of a [o]. | A picture of a [c] [o]. |
| | An image of a [o]. | An image of a [c] [o]. |
| | An image of a [o]. | An image of a [o] which is [c]. |
| | There is an image of a [o]. | There is an image of a [c] [o]. |
| | There is a photo of a [o]. | There is a photo of a [c] [o]. |
| | There is a picture of a [o]. | There is a picture of a [c] [o]. |
| | There is an image of a [o]. | There is an image of a [o] which is [c]. |
| | There is a photo of a [o]. | There is a photo of a [o] which is [c]. |
| | It is an image of a [o]. | It is an image of a [o] which is [c]. |
| | It is a photo of a [o]. | It is a photo of a [o] which is [c]. |
| | There is a [o]. | There is a [o] in [c]. |
| | There is a [o]. | There is a [o] which is [c]. |
| | Everyone knows [o]. | Everyone knows that [o] is [c]. |
| | Everyone knows [o]. | Everyone knows that [o] is [c]. |
| Shape | This is a [o]. | This is a [o] with [c] shape. |
| | There is a [o]. | There is a [c] [o]. |
| | There is a [o]. | There is a [o] which shape is [c]. |
| | It is an image of a [o]. | It is an image of a [o] which shape is [c]. |
| | There is an image of a [o]. | It is an image of a [o] which shape is [c]. |
| | There is an image of a [o]. | There is an image of a [c] [o]. |
| | There is a picture of a [o]. | There is a picture of a [c] [o]. |
| | There is a picture of a [o]. | There is an picture of a [o] which shape is [c]. |
| | There is a picture of a [o]. | There is an picture of a [c] [o]. |
| | This is a picture of a [o]. | This is a picture of a [o] has [c] shape. |
| | A picture of a [o]. | A picture of a [o] has [c] shape. |
| | An image of a [o]. | An image of a [c] [o]. |
| | A photo of a [o]. | A photo of a [c] [o]. |
| | A picture of a [o]. | A picture of a [c] [o]. |
| | [o] is of shape . | [o] is of shape [c]. |
| | The shape of [o]. | The shape of [o] can be [c]. |
| | The shape of the [o]. | The shape of the [o] is [c]. |
| Material | This is an image of a [o]. | This is an image of a [o] made of [c]. |
| | This is an image of a [o]. | This is an image of a [o] which made from [c]. |
| | This is an image of a [o]. | This is an image of a [o] which made of [c]. |
| | This is a photo of a [o]. | This is a photo of a [o] made of [c]. |
| | This is a picture of a [o]. | This is a picture of a [o] made of [c]. |
| | This is a picture of a [o]. | This is a picture of a [o] which made of [c]. |
| | It is a picture of a [o]. | It is a picture of a [o] made of [c]. |
| | A picture of a [o]. | A picture of a [o] which made from [c]. |
| | A picture of a [o]. | A picture of a [o] which made of [c]. |
| | A picture of a [o]. | A picture of a [c] [o]. |
| | There is an image of a [o]. | There is an image of a [c] [o]. |
| | There is a photo of a [o]. | There is an photo of a [c] [o]. |
| | There is a picture of a [o]. | There is an picture of a [c] [o]. |
| | An image of a [o]. | An image of a [c] [o]. |
| | A photo of a [o]. | A photo of a [c] [o]. |
| | A picture of a [o]. | A picture of a [c] [o]. |

Table 4: Templates we used for constructing anchor sentences and candidate sentences. The templates for CoDa are the same as Color.

# E   MORE EXPERIMENTAL RESULTS

Since not all models are compatible with all methods, we exclude the results of incompatible model-method combinations from the main text. The complete results are provided in Table 5. Notably, the results of Mistral$_{w/CL}$ with the verbalization-based method is 0, as this model, trained via contrastive learning, has significantly lost its ability to follow instructions, preventing it from generating reasonable responses based on prompts.

| | Model (#Inference Parameters) | CoDa | Color | Shape | Material | CFC |
|---|---|---|---|---|---|---|
| | Baselines | | | | | |
| CSM | ACCENT (440M) | 10.07 | 10.35 | -2.10 | 16.99 | 35.04 |
| | COMET-Atomic-2020-Bart (440M) | 22.91 | 26.98 | 40.44 | 25.72 | - |
| | VERA-T5-XXL (5B) | 58.93 | 45.08 | 30.31 | 33.51 | 45.81 |
| LM | RoBERTa+likelihood (355M) | 24.37 | 33.63 | 36.12 | 24.23 | 42.46 |
| | RoBERTa$_{w/CL}$+likelihood (355M) | 23.36 | 31.51 | 26.69 | 22.23 | 38.03 |
| | Mistral+verbal. (7B) | 46.64 | 38.63 | 30.46 | 36.34 | 32.06 |
| | Mistral+likelihood (7B) | 51.30 | 34.31 | 26.70 | 37.03 | 47.98 |
| | Mistral$_{w/CL}$+verbal. (7B) | 0.0 | 0.0 | 0.0 | 0.0 | 0.0 |
| | Mistral$_{w/CL}$+likelihood (7B) | 25.70 | 4.72 | 18.81 | 5.96 | 35.46 |
| | Qwen2+verbal. (7B) | 57.40 | 41.59 | 38.3 | 36.76 | 29.32 |
| | Qwen2+likelihood (7B) | 50.25 | 40.99 | 32.52 | 37.13 | 45.10 |
| | Qwen2$_{w/CL}$+verbal. (7B) | 11.12 | 15.28 | -24.21 | 0.45 | 21.39 |
| | Qwen2$_{w/CL}$+likelihood (7B) | 49.65 | 41.75 | 32.8 | 37.3 | 43.00 |
| | ComPASS | | | | | |
| LM | RoBERTa$_{w/CL}$ (355M) | 44.59 | 38.92 | 42.92 | 33.55 | 44.46 |
| | Mistral$_{w/CL}$ (7B) | 58.54 | 42.20 | 43.75 | **38.77** | **49.01** |
| | Qwen2$_{w/CL}$ (7B) | 59.16 | 44.61 | 47.51 | 38.49 | 46.41 |
| VLM | CLIP (124M) | 58.10 | 45.55 | 45.82 | 33.56 | 35.13 |
| | EVA-CLIP (695M) | **62.87** | **51.73** | **48.05** | 38.67 | 41.46 |

Table 5: Spearman's rank correlation coefficient $\rho$ between the predicted ranks of candidates and their ground-truth on CoDa, ViComTe (Color, Shape, and Material), and CFC, shown in percentage. The **best** and second best results are highlighted in bold and underlined, respectively. '+likelihood' indicates using the likelihood-based method and '+verbal.' indicates using the verbalization-based method.

