# OpenReview forum: "Estimating Commonsense Plausibility through Semantic Shifts"
_ICLR.cc/2026/Conference — ICLR 2026 Conference Withdrawn Submission_

### Official Review · Reviewer_3dXt · 2025-10-30

**Soundness:** 2
**Presentation:** 3
**Contribution:** 2
**Rating:** 2
**Confidence:** 3

**Summary:**

The paper proposes ComPaSS, a zero shot discriminative framework for commonsense plausibility estimation that ranks candidate completions by the semantic shift they induce. For each instance, the authors form an anchor sentence that omits the target detail and a candidate sentence that adds it, then score plausibility with a similarity function between their sentence embeddings p_i \propto \mathrm{sim}(r_{\text{anchor}}, r_{\text{candi}}). The approach is architecture agnostic and works with encoder LMs, decoder LLMs, and VLM text towers; anchor and candidate sentences are built from hand templates for structured triplets and with a GPT 4 transformation prompt for free form Q A. Evaluations on CoDa and ViComTe attribute ranking, plus CFC frame completion, show consistent gains over likelihood and verbalization baselines, with VLM backbones particularly strong on visually grounded attributes and contrastive pretraining crucial for performance. For example, EVA CLIP with ComPaSS attains \rho=62.87 on CoDa and \rho=51.73 on ViComTe Color, while contrastive RoBERTa improves from 24.37 to 44.59 on CoDa relative to likelihood.

**Strengths:**

•	Strong empirical results across three datasets and multiple backbones with consistent improvements over likelihood and verbalization setups, including against GPT 4 in binary comparisons
•	Clear evidence that sentence level context matters and that contrastive pretraining is the main driver of discriminative quality in the embeddings
•	Insightful analysis that VLM text towers better capture visually grounded commonsense and can mitigate reporting bias patterns seen in text only models
•	Parameter efficiency observations are useful, showing encoder scale models plus ComPaSS can rival or beat larger decoder LLMs

**Weaknesses:**

•	Conceptual novelty is limited, since scoring candidates by cosine similarity shifts between anchor and augmented sentences closely relates to prior embedding based plausibility and NLI style scoring; the paper would benefit from a direct head to head with such discriminative baselines
•	The free form pipeline relies on GPT 4 to transform Q A into sentences, which introduces closed source dependencies, output variance, and potential leakage of external knowledge not available to the backbones; reproducibility and fairness relative to baselines that do not use GPT 4 need discussion
•	Possible data leakage for VLMs is not addressed: ViComTe is derived from Visual Genome, which is widely used in web corpora for VLM pretraining; without deduplication checks, improvements may partially reflect overlap
•	Baseline tuning appears uneven. For example, contrastively trained Mistral loses instruction following and scores 0 with verbalization, which weakens the generative comparison; likelihood baselines may be sensitive to normalization and prompt formatting choices that are not fully explored
•	Statistical rigor is thin. CFC uses only the 55 example validation set, confidence intervals or significance tests are not reported, and sensitivity to the number of GPT 4 samples or template choices is only partially ablated
•	The similarity signal may be confounded by lexical overlap and template artifacts. More analysis is needed to show robustness to paraphrase, negation, and longer contexts

**Questions:**

•	Are images ever used at inference time or is the method exclusively using the text tower of VLMs? If only text is used, please clarify why the text tower of CLIP style models should outperform specialized text encoders beyond contrastive benefits, and provide a control that matches training size
•	Can the authors audit potential training data overlap between CLIP style pretraining corpora and ViComTe Visual Genome derived items, and re run with a filtered checkpoint or an overlap free subset?
•	How sensitive are results to the similarity function and pooling strategy? Please add cosine vs dot product and CLS vs EOS pooling ablations for at least one backbone
•	For the GPT 4 transformation step, can you release the generated sentence pairs, report stochasticity across runs, and show that replacing GPT 4 with a small open source rewritier leaves conclusions unchanged?
•	Could you compare with a strong discriminative baseline such as NLI based scoring or energy based reranking that does not use generative likelihoods, to isolate the benefit of the semantic shift idea?
•	Please provide confidence intervals or paired significance tests for the main tables, especially for CFC where n=55
•	Do lexical overlap controls change outcomes? For example, mask or paraphrase color words in templates and test whether rank correlations remain high

---

### Official Review · Reviewer_AZ9z · 2025-10-31

**Soundness:** 2
**Presentation:** 3
**Contribution:** 2
**Rating:** 4
**Confidence:** 3

**Summary:**

This paper introduces ComPaSS (Commonsense Plausibility through Semantic Shifts), a discriminative framework for evaluating commonsense plausibility in language models. The core idea is to measure semantic shifts when augmenting sentences with commonsense-related information: plausible augmentations cause minimal semantic shifts, while implausible ones result in substantial deviations. The authors evaluate ComPaSS on two task types: attribute value ranking (structured triplet inputs) and commonsense frame completion (free-form question-answer pairs). Experiments across various language models and vision-language models demonstrate that ComPaSS consistently outperforms likelihood-based and verbalization-based baselines, with particular advantages when using VLMs for vision-grounded commonsense and models with contrastive pre-training.

**Strengths:**

Novel Discriminative Perspective for CSPE: The paper successfully reframes commonsense plausibility estimation as a discriminative task rather than a generative one, which is both intuitive and theoretically well-motivated. The insight that plausible commonsense knowledge should integrate seamlessly (minimal semantic shift) into context is elegant and aligns with how humans process implicit information.

Comprehensive Experimental Evaluation: The experimental design is thorough, covering multiple dimensions: different model architectures (encoders and decoders), different modalities (LMs and VLMs), various model sizes (from 124M to 7B parameters), and different types of commonsense knowledge (visual attributes versus general knowledge). The inclusion of both structured triplet tasks and free-form completion tasks demonstrates the method's generalizability across input formats.

Strong Empirical Results: ComPaSS achieves consistently superior performance across all evaluated tasks and model backbones. The improvements are substantial, with EVA-CLIP achieving 62.87% on CoDa compared to 58.93% for the strongest baseline (VERA-T5). The method even outperforms GPT-4 on several benchmarks, which is noteworthy given GPT-4's scale and capabilities.

**Weaknesses:**

Limited Theoretical Justification: While the intuition behind ComPaSS is appealing, the paper lacks rigorous theoretical justification for why semantic shift should correlate with commonsense plausibility. Under what formal conditions does minimal semantic shift imply plausibility? Are there cases where this assumption breaks down? The paper would benefit from a more formal analysis of when and why this heuristic works, perhaps drawing on linguistic theory or formal semantics.

Heavy Reliance on GPT-4 for Sentence Construction: For free-form question-answer tasks, ComPaSS uses GPT-4 to construct anchor and candidate sentences. This introduces several concerns: First, it creates a dependency on a closed-source, expensive API, limiting reproducibility and scalability. Second, the quality of sentence construction directly impacts ComPaSS performance, yet this component is not systematically evaluated.

Insufficient Analysis of Failure Cases: The paper does not systematically analyze when ComPaSS fails. What types of commonsense reasoning does semantic shift measurement struggle with?

Limited Evaluation on CFC Dataset: The Commonsense Frame Completion evaluation uses only 55 validation examples because the test set is not public. This small sample size raises concerns about statistical significance and generalizability. The paper should report confidence intervals or conduct bootstrap analysis to assess result reliability. Additionally, seeking access to the full test set or evaluating on alternative free-form commonsense reasoning datasets would strengthen claims about generalizability beyond structured attribute tasks.

Reporting Bias Analysis Lacks Depth: The paper attributes VLM superiority to overcoming reporting bias, using the "black sheep" example as illustration. However, this analysis remains somewhat superficial. A more rigorous approach would involve: quantifying reporting bias systematically across multiple objects and attributes, comparing frequency statistics in text corpora versus visual datasets, and demonstrating that performance gains specifically correlate with high-bias examples. The current treatment, while intuitive, lacks empirical depth.

**Questions:**

1. Can the authors provide formal conditions under which semantic shift reliably indicates implausibility? Are there linguistic or cognitive science theories that support this framework? What happens when multiple commonsense interpretations are equally plausible but semantically distant?

2. How sensitive is ComPaSS to the quality of sentence construction? Can the authors ablate the GPT-4 dependency by comparing against template-based sentence construction or other open-source LLMs? What percentage of the performance gain comes from GPT-4's sophisticated sentence generation versus the semantic shift measurement itself?

3. What are the computational costs of ComPaSS compared to likelihood-based methods? For ranking K candidates, ComPaSS requires encoding 2K sentences (anchor + each candidate), whereas likelihood methods may compute a single forward pass. How does this scale to scenarios with many candidates?

4. Commonsense knowledge can vary across cultures and languages. How well does ComPaSS generalize to non-English languages or culture-specific commonsense?

---

### Official Review · Reviewer_LtEJ · 2025-11-01

**Soundness:** 2
**Presentation:** 2
**Contribution:** 2
**Rating:** 2
**Confidence:** 3

**Summary:**

This paper introduces ComPaSS, a zero-shot approach for estimating commonsense plausibility by leveraging semantic shifts in sentence representations. The method is evaluated on multiple tasks and datasets, demonstrating its flexibility across different model architectures, including both encoder-only and decoder-only models. The experiments highlight the importance of sentence-level representations and contrastive learning for achieving strong performance in commonsense reasoning tasks.

**Strengths:**

1. **Generalizability Across Model Architectures.**
    ComPaSS is designed to be architecture-aware and can be flexibly applied to both encoder-only and decoder-only models, demonstrating broad applicability.
2. **Zero-Shot Capability.**
    The method does not require task-specific fine-tuning, enabling zero-shot commonsense plausibility estimation, which is valuable for practical deployment.

**Weaknesses:**

1. The paper primarily experiments with RoBERTa, CLIP-series, Mistral, and Qwen2 models. While these cover both encoder-only and decoder-only backbones, the evaluation does not include more recent state-of-the-art models such as Llama3 or Qwen3, which feature advanced scaling and instruction-following capabilities. This omission may limit the generalizability and contemporary relevance of the findings.
2. Although the paper discusses the parameter efficiency of encoder-only models compared to larger decoder-only models, it lacks a systematic comparison of ComPaSS performance across different parameter scales within the same model family (e.g., 1B ~ 13B variants). Such an analysis would help clarify the scalability and robustness of the method under varying model capacities.
3. According to the experimental setup, ComPaSS is mainly evaluated on models that have undergone contrastive learning pre-training (e.g., sup-SimCSE-RoBERTa-Large, E5-Mistral-7B-Instruct, gte-Qwen2-7B-instruct). There is little or no reporting of results on base models without contrastive pre-training. The paper mentions that applying ComPaSS to vanilla RoBERTa leads to performance degradation, indicating the method's reliance on high-quality semantic representations typically enhanced by contrastive learning. However, a more thorough ablation or discussion on the applicability to base models would strengthen the claims.

**Questions:**

1. In Figure 4, the label "get-Qwen2" appears. Is this a typographical error, and should it be "gte-Qwen2"?
2. Could the authors specify the exact versions of the models used in experiments? For example, does "Mistral-7B-Instruct" correspond to "Mistral-7B-Instruct-v0.1" or another version?
3. The paper reports that ComPaSS performs poorly on vanilla RoBERTa due to weaker representation capabilities, whereas contrastive learning significantly improves performance. Is the lack of results for ComPaSS on base (non-contrastive) models intentional, and does this imply that ComPaSS is fundamentally unsuitable for base LMs without contrastive pre-training? Would it be possible to clarify if contrastive learning is a prerequisite for the effective use of ComPaSS?

---

### Official Review · Reviewer_AMZ6 · 2025-11-03

**Soundness:** 2
**Presentation:** 2
**Contribution:** 2
**Rating:** 4
**Confidence:** 4

**Summary:**

This paper introduces ComPaSS (Commonsense Plausibility through Semantic Shifts), a novel discriminative framework for fine-grained commonsense plausibility estimation. The core innovation lies in quantifying plausibility by measuring semantic shifts when augmenting sentences with commonsense information - plausible augmentations induce minimal semantic shifts, while implausible ones cause substantial deviations. The methodology operates in three stages: (1) constructing anchor sentences (base context without explicit commonsense) and candidate sentences (augmented with commonsense information), (2) encoding both using pre-trained language models (LMs) or vision-language models (VLMs) to obtain semantic representations, and (3) ranking candidates by semantic similarity between anchor and candidate representations, where higher similarity indicates greater plausibility. The authors evaluate ComPaSS on two task types: attribute value ranking using structured triplet inputs (CoDa and ViComTe datasets covering color, shape, and material attributes) and commonsense frame completion using free-form question-answer pairs (CFC dataset). Experiments span multiple model architectures including RoBERTa, Mistral-7B, Qwen2-7B, CLIP, and EVA-CLIP, with and without contrastive learning pre-training.

**Strengths:**

1. The paper is generally well-written with clear motivation and methodology.

2. The discriminative perspective for commonsense plausibility estimation is both novel and well-motivated. The insight that commonsense knowledge is often implicit in communication, and that plausible augmentations should integrate with minimal semantic disruption, provides a foundation for commonsense plausibility.

3. The extensive comparison across diverse model architectures (encoder-only, decoder-only, unimodal, and multimodal) with varying parameter scales (124M to 7B) provides strong empirical support for the approach's architecture-agnostic effectiveness.

**Weaknesses:**

1. While the application is novel, the fundamental technique of measuring semantic similarity via embedding cosine similarity is well-established. The paper would benefit from deeper theoretical analysis of why semantic shift specifically captures commonsense plausibility better than alternatives. Refer the following papers: https://dl.acm.org/doi/10.1145/3672393, https://arxiv.org/abs/2304.01666

2. The paper lacks systematic analysis of when and why ComPaSS fails. While Table 2 shows vanilla RoBERTa performs poorly with ComPaSS, there is no investigation into: (1) which types of commonsense knowledge are most challenging (temporal, causal, social, physical?), (2) error analysis comparing failure modes of ComPaSS versus baselines.

3. Several relevant baselines are missing: (1) retrieval-augmented approaches that use external commonsense knowledge bases (https://aclanthology.org/2022.emnlp-main.294.pdf), (2) recent prompting techniques like chain-of-thought or self-consistency for LLMs (https://arxiv.org/abs/2201.11903, https://arxiv.org/abs/2203.11171)

4. Minor weakness: Despite overall clarity, some sections suffer from: (1) inconsistent notation (e.g., switching between c for context and c for color), (2) vague statements like "significantly better" without quantification, (3) overclaiming (e.g., "first comprehensive study" when similar ideas exist in semantic shift detection literature)

**Questions:**

1. Why does semantic shift specifically capture commonsense plausibility? Can you provide theoretical or empirical analysis distinguishing semantic shift from other embedding-based measures (e.g., embedding norm changes or divergence measures)? Have you tested alternative similarity metrics beyond cosine similarity?

2. Have you ensured fair comparison against baselines by: (a) tuning prompt formatting and few-shot examples for verbalization methods? (b) comparing against more sophisticated prompting techniques (chain-of-thought, self-consistency)?

---

### Note · Authors · 2025-12-01

I have read and agree with the venue's withdrawal policy on behalf of myself and my co-authors.